# Chronic Heat Stress Induces Acute Phase Responses and Serum Metabolome Changes in Finishing Pigs

**DOI:** 10.3390/ani9070395

**Published:** 2019-06-28

**Authors:** Yanjun Cui, Chong Wang, Yue Hao, Xianhong Gu, Haifeng Wang

**Affiliations:** 1Institute of Animal Nutrition, College of Animal Science and Technology, Zhejiang A & F University, Lin’an 311300, China; 2Key Laboratory of Applied Technology on Green-Eco-Healthy Animal Husbandry of Zhejiang Province, Hangzhou 311300, China; 3State Key Laboratory of Animal Nutrition, Institute of Animal Sciences, Chinese Academy of Agricultural Sciences, Beijing 100081, China; 4College of Animal Science, MOE Key Laboratory of Molecular Animal Nutrition, Zhejiang University, Hangzhou 310058, China

**Keywords:** finishing pigs, chronic heat stress, acute-phase proteins, metabolomics

## Abstract

**Simple Summary:**

There is limited information on the serum metabolome of heat-stressed finishing pigs. Our results indicated that heat stress led to oxidative stress and acute phase response. Pigs chronically exposed to high ambient temperature were in negative energy balance status. Three gut microbiome-derived metabolites (fluorine, lyxose 1, and D-galacturonic) were likely to be biomarkers for monitoring animal health.

**Abstract:**

Heat stress (HS) is a main environmental challenge affecting the animal welfare and production efficiency in pig industry. In recent years, numerous reports have studied the alterations in gene expressions and protein profiles in heat-stressed pigs. However, the use of metabolome to unravel adaptive mechanisms of finishing pig in response to chronic HS have not yet been elucidated. We aimed to investigate the effects of chronic HS on serum metabolome in finishing pigs, and to identify the biomarkers of heat stress. Pigs (n = 8 per treatment) were exposed to either thermal neutral (TN; 22 °C) or heat stress (HS, 30 °C) conditions for three weeks. Serum metabonomics of TN- and HS-treated pigs were compared using the GC-MS approach. Metabonomics analysis revealed that twenty-four metabolites had significantly different levels in TN compared to HS (variable importance in the projection values >1 and *p* < 0.05). These metabolites are involved in carbohydrate, amino acid, fatty acid, amines metabolism, and gut microbiome-derived metabolism. Three serum monoses (glucose, mannose 2, and galactose) and 6-phosphogluconic acid were decreased, indicating insufficient source of fuel for energy supply, resulting in negative energy balance (NEB) in heat-stressed pigs. Increased levels of non-esterified fatty acid (myristic acid, palmitic acid, and linoleic acid) and short-chain fatty acids (3-hydroxybutanoic acid and maleic acid) suggested fat decomposition compensating for energy shortage, which was an adaptive response to NEB. Increased concentrations of fluorine, lyxose 1, and D-galacturonic acid were significantly correlated with the levels of acute phase proteins (HP, LBP, α2-HSG, and Lysozyme), suggesting acute phase response in HS-stressed pigs. These metabolites are expected to be novel biomarkers of chronic HS in pigs, yet the use of which awaits further validation.

## 1. Introduction

Heat stress (HS) is a main environmental stressor challenging the animal welfare, production efficiency, and health status in livestock industry [1]. In comparison to other livestock animals, pigs are more susceptible to high ambient temperatures as they have higher metabolic heat production and lower ability to heat dissipation due to limited sweat glands and substantial adipose deposition [1]. HS gives rise to reduced performance, feed efficiency, and meat quality, all of which in turn generates economic losses.

HS is suggested to be a contributor of oxidative stress in livestock animals [2]. Previous studies have concluded that chronic exposure to high ambient temperatures accelerates reactive oxygen species (ROS) generation, which could induce cytotoxicity, such as lipid peroxidation and oxidative damages to proteins [3]. Animals develop a defense system of antioxidant enzymes against ROS, including superoxide dismutase (SOD), catalase (CAT) and glutathione peroxidase (GSH-PX). The enzymes are considered markers of redox homeostasis and monitoring their activities can provide the objective measure of stress extent.

HS is believed to induce inflammatory and acute phase response (APR) in humans, pigs, and dairy cows [4,5,6,7]. Our and other studies have suggested that HS could compromise intestine barrier in pigs [5,7], facilitating endotoxin translocation of bacteria and endotoxins into blood, which in turn initiates an inflammatory response and production of cytokines such as interleukin (IL)-6, IL-1β, and tumor necrosis factor-α (TNF-α). The APR is a core part of innate immune system, which can be triggered by various challenges including stress, inflammation, and infection [8]. Acute phase proteins (APP) are a group of blood proteins mainly generated by hepatocytes and involved in APR. The APPs, such as haptoglobin (HP), serum amyloid A (SAA), and C-reactive protein (CRP), are all considered as positive acute phase reactants; retinol-binding protein (RBP), serum albumin, and alpha-2-HS-Glycoprotein (α2HSG) as negative acute phase reactants. HS was found to increase HP production in blood and liver of growing and finishing pigs, respectively [5,9,10]. Chronic HS in finishing pigs down-regulated the protein expression of albumin, α2HSG, and RBP in the intestinal mucosa [7]. APPs, such as SAA, HP, and CRP were recognized markers of inflammation and have also been proposed as indicators of stress in cattle [11] and pigs [3]. Nevertheless, a global approach to the whole organism adaptation response to HS conditions has not been carried out and it may yield information about new biomarkers that help to evaluate animal welfare in an objective manner.

Recent studies by our research group have reported that exposure to chronic HS (30 °C) for 21 d in finishing pigs resulted in increase in plasma cortisol and decrease in plasma free triiodothyronine and growth hormone [7,12], which reduced the basal metabolic rate and change the expression of proteins involving in energy and nutrient metabolites. However, the global metabolic changes of heat-exposed pigs remain largely unknown. Metabolomics facilitate the identification of thousands of metabolites, providing the tools for exploring the molecular mechanisms of human and animal response to environmental stress and nutritional imbalance. Based on the metabolomics analysis, metalolites are expected to be potential biomarkers for monitoring animal stress and welfare.

## 2. Materials and Methods

### 2.1. Animals and Experimental Design

The experiment was performed in accordance with the guidelines of Beijing Animal Ethics Committee and received prior approval from the Chinese Academy of Agricultural Sciences Animal Care and Use Committee. Sixteen castrated male crossbreeds of Landrace Yorkshire sows and Duroc boars (body weight = 79.00 ± 1.50 kg) pigs were randomly assigned to either thermal neutral (TN) (22 °C) or chronic heat stress (HS) (30 °C) group (Eight pigs per treatment group). Four pigs from one group were housed in an artificial climate chamber (2.1 × 4.8 m^2^, luminance 100 lx, photoperiod 14 h light, humidity 55 ± 5%) as described by our previous studies [7]. Sixteen pigs were allocated to four the climate chambers. All pigs were ad libitum fed with standard diet according to the NRC (1998) recommendations. The feed contained no antibiotics (Appendix A).

Before the official trial, the animals were acclimated to the artificial climate chamber at 22 °C for 7 days. To minimize damage caused by high temperature exposure in the HS group, the artificial temperature climate of the chamber was gradually elevated and maintained at 27 °C on d 1, and then raised to 28 °C on d 2. The treatment was lasted for 3 weeks after the temperature of HS group was reached and maintained at 30 °C.

Right after the 3 weeks’ treatment and prior to sacrifice, venous blood was immediately collected from the jugular vein using venipuncture and centrifuged at 1500× *g* at 4 °C for 10 min to obtain serum. The serum samples were subsequently transferred into 1.5 mL sterile tubes and stored at −80 °C until later assay.

The whole blood erythrocyte, hemoglobin (Hb), hematocrit (Hct) were analyzed by using stat profile PHOX reagent pack a calibration cartridge-B by ABG blood gas analyzer (NOVA Biomedicals, MA, USA). Serum Anti-oxidant enzyme activities of superoxide dismutase (SOD), catalase (CAT), and glutathione peroxidase (GSH-PX) and malondialdehyde (MDA) level in serum were determined using the corresponding assay kits (Nanjing Jiancheng Bioengineering Institute, Nanjing, China) according to the manufacturer’s instructions.

Serum haptoglobin (HP), alpha-2-HS-Glycoprotein (α2HSG), LPS-blinding protein (LBP), and lysozyme were measured using corresponding porcine-specific ELISAs according to the manufacturer’s instructions (Beijing Sino-UK Institute of Biological Technology, Beijing, China).

### 2.2. Sample Preparation

The serum samples were thawed at room temperature and prepared as previously described [13], but with some modifications. Briefly, each serum sample (80 μL) was mixed with 10 μL 2-chloro-L-phenylalanine (0.3 mg/mL in methanol; Hengchuang Biotech) as an internal quantitative standard in tubes, and vortexed for 10 s. Subsequently, 240 μL of an ice-cold methanol-chloroform (V_methanol_:V_chloroform_ = 2:1) was added to each tube, which was then vortexed for 1 min, ultrasonicated at 25 °C for 5 min, and chilled at −20 °C for 10 min. The extracts were centrifuged at 12,000 rpm at 4 °C for 15 min. A quality control (QC) sample was prepared by mixing aliquots of all the samples into a pooled sample. The supernatants were transferred to a GC-TOF-MS glass vials, dried in a vacuum concentrator at room temperature. 80 μL methoxylamine hydrochloride (15 mg/mL in pyridine) was added into the dried metabolite extracts, vortexed vigorously for 2 min, and then incubated at 37 °C. 80 μL N, O-bis (trimethylsilyl) trifluoroacetamide (containing 1% TCMS, v/v) and 20 μL n-hexane (CNW Technologies GmbH) was added to each mixture, and the mixtures were incubated at 70 °C for 1 h. The samples were kept at room temperature for 30 min before GC-MS analysis.

### 2.3. GC-TOF-MS Analysis

Based on a previous study, GC-TOF-MS analysis was conducted using an Agilent 7890 gas chromatograph system linked to a Pegasus HT time-of-flight mass spectrometer (LECO, St. Joseph, MI, USA). A DB-5MS fused-silica capillary column (30 m × 0.25 mm × 0.25 μm; Agilent J & W Scientific) was used to separate the derivatives. He (>99.999%) was used as the carrier gas, at a constant flow rate of 1 mL/min, and the injector temperature was maintained at 260 °C. The injection volume was 1 μL by splitless mode, and the solvent delay time was 5 min. The column temperature was increased from 50 to 125 °C at a rate of 15 °C/min, increased to 210 °C at a rate of 5 °C/min, increased to 270 °C at a rate of 10 °C/min, increased to 305 °C at a rate of 20 °C/min, and then kept at 305 °C for 5 min. The MS quadrupole temperature was set to 150 °C, and the ion source temperature was set to 230 °C. The collision energy was 70 eV. Mass data were acquired using a full-scan mode (*m*/*z* 50–450). The QC sample was injected at regular intervals throughout the analytical run, so as to assess repeatability.

### 2.4. Data Processing and Analyses

Statistical analyses of serum biochemical parameters were performed using SAS version 8.2 software (SAS Institute, Cary, NC, USA). Data were expressed as means and standard deviations. The Student’s *t*-test was used for statistical analysis and a difference at *p* < 0.05 was considered statistically significant. ChemStation (version E.02.02.1431; Agilent) was used to convert the file format of the raw data to common data format, and ChromaTOF (version 4.34; LECO) was used to analyze the data. The metabolites were identified qualitatively from the NIST and Fiehn databases. Normalized data were imported into the SIMCA software package (14.0; Umetrics, Umea, Sweden) orthogonal projections to latent structures discriminant analyses (OPLS-DA) so as to visualize differences in the metabolism between experimental groups. Variable importance in the projection (VIP) ranks the overall contribution of each variable to the OPLS-DA model, so variables with VIP values of >1 were considered relevant for group discrimination. Differential metabolites were selected based on the combination of statistically significant VIP values and *p* values from a two-tailed Student’s *t* test of the normalised peak areas, that is, metabolites with VIP values of >1 and *p* values of <0.05, and were further identified and performed functional analysis by searching the Kyoto Encyclopedia of Genes and Genomes (KEGG). Hierarchical clustering, *k*-Mean clustering and heat map generation of proteins were performed through Multi Experiment Viewer (MeV). Pearson’s correlation coefficients were carried out to determine correlations between acute phase proteins and differential metabolites of the TN and HS groups.

## 3. Results

### 3.1. Blood Parameter

The effects of chronic HS on erythrocyte, hemoglobin (Hb), hematocrit (Hct), mean corpuscular volume (MCV) and mean corpuscular hemoglobin (MCH) in whole blood are shown in Table 1. Compared to the TN group, HS exposure significantly decreased their values (*p* < 0.05).

### 3.2. Antioxidant and Oxidative Biomarkers

The antioxidant parameters including activity of SOD, CAT, and GSH-PX as well as MDA concentration in serum are shown in Figure 1. Compared to the TN group, pigs exposed to HS condition for 3 weeks had a significant decrease in activities of CAT (4.65 vs. 3.65 U/mL, *p* = 0.042) and GSH-PX (838.68 vs. 701.21 U/mL, *p* = 0.002). SOD activity and MDA concentration were not significantly affected by HS (*p* > 0.05).

### 3.3. Acute Phase Response

Compared to TN pigs, serum HP concentrations increased by 19.0% (Figure 2A, *p* = 0.007) whereas that of α2HGS decreased by 17.6% (Figure 2B, *p* = 0.003) in the HS pigs. Serum LBP concentrations increased by 12.3% (Figure 2C, *p* = 0.05) due to HS. Serum lysozyme activity was decreased by 23.0% (Figure 2D, *p* < 0.001) in heat-stressed pigs compare with the TN pigs.

### 3.4. Metabolic Profiles and Orthogonal Supervised Pattern Recognition

Metabolomic analysis was carried out using GC-MS approach to determine the metabolite profile of serum samples collected from HS and TN pigs. In total, 473 valid peaks and 214 metabolites were identified in the serum samples. The orthogonal partial least squares discriminate analysis (OPLS-DA) plots of the metabolomic data showed a clear separation between the HS and TN groups, without any overlap (Figure 3A). The parameters of the HS vs. NC model, *R2Y* (cum) = 0.886 and *Q2* (cum) = −0.561, indicated that samples (data records) fit the established discriminant mathematic model. To avoid model overfitting, cross-validation across three components with 999 random permutation tests was carried out and generated intercepts of 0.106 < *R*^2^ < 0.415 and −0.266 < Q^2^ < −0.125 (Figure 3B). The OPLS-DA models indicate the significant alterations of metabolic profile in serum of finishing pigs exposed to HS.

### 3.5. Metabolic Changes

In total, twenty-four metabolites with significantly different levels (VIP > 1 and *p* < 0.05) in the HS groups were identified, and the differences were expressed as fold-change values. Of them, fifteen and nine had higher and lower concentrations in the HS group than in the NC group, respectively (Table 2). An overview of the hierarchical clustering analysis of metabolites with significantly different levels revealed expression pattern (Figure 4).

HS induced increase in the levels of fluorine, lyxose 1, and D-galacturonic acid, all of which are associated with gut microbiome-derived metabolites. Compared with the TN group, the levels of 3-Hexenedioic acid, myristic Acid, palmitic acid, maleic acid, tartaric acid and 3-Aminoisobutyric acid 1 increase and that of arachidonic acid decreased in HS group, and these metabolites were involved in fatty acid metabolism. HS-induced alternations in carbohydrate-related metabolites were observed for glucose 2, mannose 2, lactose 1, 3,6-Anhydro-D-galactose, 6-phosphogluconic acid, ribonic acid, gamma-lactone, glucoheptonic acid 1, and D-galacturonic acid. HS increased the levels of lysine and N-Methyl-DL-alanine, both of which are associated with amino acid metabolism. Hydroxylamine, N-Acetyl-5-hydroxytryptamine 2, 3-methylcatechol, and indole-3-acetamide 4 are associated with amines metabolism. In comparison with TN group, the levels of N-Acetyl-5-hydroxytryptamine 2, 3-methylcatechol, and indole-3-acetamide were decreased and that of hydroxylamine was increased in HS group.

### 3.6. Correlation Analysis

Table 3 lists the correlation coefficients between stress biomarker (HP, LBP, LYS, and α2HSG) and candidate metabolite levels in serum, respectively. The HP concentration was positively correlated with fluorine, 6 phosphogluconic acid, Arachidonic acid, Lyxose 1, D-galacturonic acid 1, N-Methyl-DL-alanine, and 3, 6-Anhydro-D-Galactose 3 and negatively correlated with Mannose 2. There was positive correction between LBP and Tartaric acid ribonic acid, gamma-lactone, Lyxose 1, D-galacturonic acid 1, 2-hydroxybutanoic acid, and palmitic acid and negative correction between LBP and Mannose 2, glucose 2, 3-Aminoisobutyric acid 1, Lysine, and maleic acid. The LYS activity was positively correlated with N-Acetyl-5-Hydroxytryptamine 2, indole-3-acetamide 4, tartaric acid, hydroxylamine, glucoheptonic acid 1, and myristic acid and negatively correlated with Mannose 2, 3-Aminoisobutyric acid 1. α2HSG concentration was positively correlated with ribonic acid, gamma-lactone, lyxose 1, D-galacturonic acid 1, 2-hydroxybutanoic acid, and palmitic acid and negatively correlated with Mannose 2, glucose 2, 3-Aminoisobutyric acid 1, and maleic acid.

## 4. Discussion

Our previous studies have demonstrated that chronic heat stress model for finishing pigs is established [7,12]. Based on this model, we used a metabolic approach to reveal the mechanism underlying the adaption of finishing pigs to chronic HS and identify to the novel biomarkers in serum monitoring stress. Serum is an optimal target as it is readily available with minimal stress for animals and is the prime carrier of various metabolites whose levels are susceptible to HS [14].

### 4.1. Blood Physiology Response of Finishing Pigs to Chronic Heat Stress

Blood plays a critical role in thermoregulation and is sensitive to heat stress exposure. Previous studies have demonstrated that HS could decrease blood erythrocytes, hemoglobin (Hb), and hematocrit (Hct) levels in various farm animals, such as goats [15], chickens [16], and cows [17]. Consistently, in the current study, erythrocyte, Hb and Hct levels were decreased in the heat-stressed pigs. These reductions might be attributed to the increased damage to the erythrocyte attacked by free radicals and inadequate nutrient availability for hemoglobin synthesis caused by decreased feed intake [18]. In support of this, decreased activities of CAT and GSH-PX in serum of finishing pigs were observed, which are associated with the production of reactive oxygen species (ROS) resulting in oxidative stress. This is consistent with the findings that a reduction in plasma antioxidant activity was observed in heat-stressed Holstein cows [19]. Similarly, a latest study reported that seasonal heat stress induced the Nrf2-mediated oxidative stress response in dairy cows [20].

Heat-stressed livestock have an increased acute phase response. HP, a positive APR, reduces the oxidative damage associated with hemolysis by binding free HP [21]. LBP is also a positive APR, which binds LPS, subsequently presenting them to CD14 to initiate innate immunity [22]. α2-HSG is a negative APR, and its serum levels decrease significantly in response to inflammation and/or infection [23]. Acute phase proteins, as part of the innate immune response, are predominantly synthesized in liver on stimulation of pro-inflammatory cytokines [24]. In response to chronic HS, positively- and negatively-responding APPs demonstrated an increase and decrease in concentrations, respectively. Our result for HP is coherent with the finding in growing pigs exposed to 7 d of HS [5]. Moreover, the alteration of serum APPs in concentrations is in line with that of intestine and liver [7]. Zachut et al. reported that seasonal heat stress induced the acute phase response in dairy cows [20]. Lysozyme, as a vital non-specific immune factor, has antibacterial, anti-inflammatory, and anti-viral functions, acting as responsibility of body’s defense [25]. The present study demonstrated that chronic HS significantly decreased the lysozyme activity. Pearce et al. founded that lysozyme activity was reduced in jejunum of growing pigs due to chronic HS (35 °C for 7 d) [5], but no effect was found in blood during acute and short-term HS (37 °C for <6 h) [26]. HP, LBP, α2-HSG, and lysozyme are expected to be the potential biomarker monitoring moderate heat stress of pigs and assessing animal wellbeing.

### 4.2. Serum Metabonomic Response of Finishing Pigs to Chronic Heat Stress

Metabolites are considered as both the building blocks of animal development and the regulators of animal health [27]. Metabolites in blood are reported to be highly responsive to environmental stress [28]. In the present study, twenty four metabolites whose concentrations were significantly affected by chronic HS were identified using the GC-MS-based metabonomic approach. These metabolites could be used as candidate biomarkers diagnosing the chronic HS. These potential markers are involved in five pathways including, carbohydrate, amino acid, fatty acid, amines metabolism, and gut microbiome-derived metabolism.

In present study, circulating glucose concentrations were decreased in the serum of chronic-heat-stressed pigs, which is in line with the findings in Holstein bull calves [29], cows [30], sheep [31]. However, other studies in poultry [32] and pigs [33] have demonstrated rises in blood glucose levels during acute HS. These results indicate that alterations in circulating glucose differ, which is dependent on animal species, duration, and severity of HS. Although heat-stressed animals mainly rely on glucose to meet energy needs, mannose 2 and galactose are alternative source of fuel [34]. Given decrease in feed intake and blood glucose, mannose 2 and galactose in our present study, the utilization of these monose to meet growth is declined. This indicates that chronic HS make the finishing pigs being negative energy balance (NEB), which is in agreement with the reports in cows [30,35] and broilers [36]. Glucose, mannose 2 and galactose can directly transform into 6-phosphogluconic acid which is a rate-limiting metabolite for the glycolysis pathway (Figure 5). Hence, NEB is in partly attributed to decrease in concentration of 6-phosphogluconic acid. In addition, lysine and N-Methyl-DL-alanine were higher in HS group. These protein amino acid could provide precursors for glucose production via deamination and gluconeogenesis so as to compensate for energy shortage because of decreased glucose, which is similar to the finding of Lu et al. [36].

Mobilization of body fat may lead to the formation of ketone bodies such as acetoacetate, acetone, and 3-hydroxybutanoic acid (BHBA), which can be used for energy supply. The increases in the concentrations of these metabolites in the HS group might be adaptive responses to reduced energy intake and NEB status. The concentration of BHBA was higher in the HS than in the TN groups. These results are supported by the findings of Tian et al. who reported that the activity of 3-hydroxybutyrate dehydrogenase in blood of dairy cow was increased due to HS [30]. In addition, maleic acid can directly transform into the fumaric acid involved in the tricarboxylic acid (TCA) cycle (Figure 5). Increased levels of maleic acid in HS group suggested that heat-stressed pigs reinforced their energy metabolism in order to adapt HS. Our and other studies have demonstrated that HS could increase the circulating concentrations of cortisol and epinephrine, both of which are considered as catabolic signals and usually accelerated lipolysis [37]. During this process, the newly produced free fatty acids undergo β-oxidation for energy supply to meet the demands of the body triglyceride mobilization. We found that the concentrations of serum non-esterified fatty acid (NEFA), including myristic acid, palmitic acid, and linoleic acid were significantly increased in the HS group, which is in line with the findings in dairy cows [30]. These results could be supported by our previous finding of decreased intramuscular fat proportion due to HS [38], suggesting stronger lipid decomposition for energy generation. A schematic presentation of some changed metabolic pathways is shown in Figure 5.

Fluorene, lyxose 1, and D-galacturonic acid are gut microbiome-derived metabolites, all of which were up-regulated in serum of heat-stressed pigs. Consistent with these results, our previous study demonstrated that d-lactate, a biomarker of leaky guts, was increased in serum due to HS [7]. These results suggested HS-induced permeability of intestine, which facilities the translocation of bacterial metabolites into circulation, leading to systemic inflammation [39]. This is supported by acute phase response occurring in HS pigs. In addition, concentrations of fluorine, lyxose 1, and D-galacturonic acid were significantly correlated with the concentrations of acute phase proteins (HP, LBP, and α2-HSG), respectively, suggesting that these metabolites are expected to be biomarker for chronic HS response in finishing pigs.

## 5. Conclusions

Finishing pigs exposed to a chronic heat load for 3 weeks had measurable impacts on serum metabolites involved in carbohydrate, amino acid, fatty acid, amines metabolism and gut microbiome-derived metabolism. HS resulted in oxidative stress and acute phase response. The chronic heat-stressed finishing pigs was in NEB. Increased fat mobilization and short-chain fatty acid for energy supply is part of the regulatory adaptations to chronic HS. Although these responses might be beneficial for the animal’s survival, it could still has restricting effect production of livestock. In addition, the present metabonomics analysis revealed several novel biomarkers of chronic HS in serum of finishing pigs, and further research is warranted to validate their use. Taken together, these results provided information on the mechanisms of chronic HS responses so as to develop nutritional intervention strategies to minimize the detrimental effects under heat exposure.

## Figures and Tables

**Figure 1 animals-09-00395-f001:**
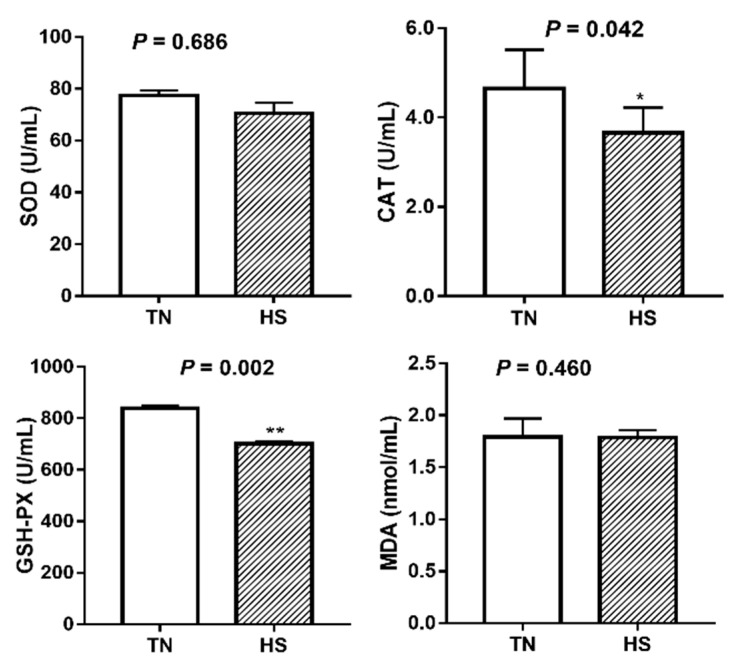
The effects of three weeks of thermal neutral (TN) or heat stress (HS) conditions on activities of SOD, CAT, GSH-PX, and MDA level in the serum of finishing pigs. Data are mean ± SD; n = 8 for each group. * *p* < 0.05 and ** *p* < 0.01 before vs. after heat stress.

**Figure 2 animals-09-00395-f002:**
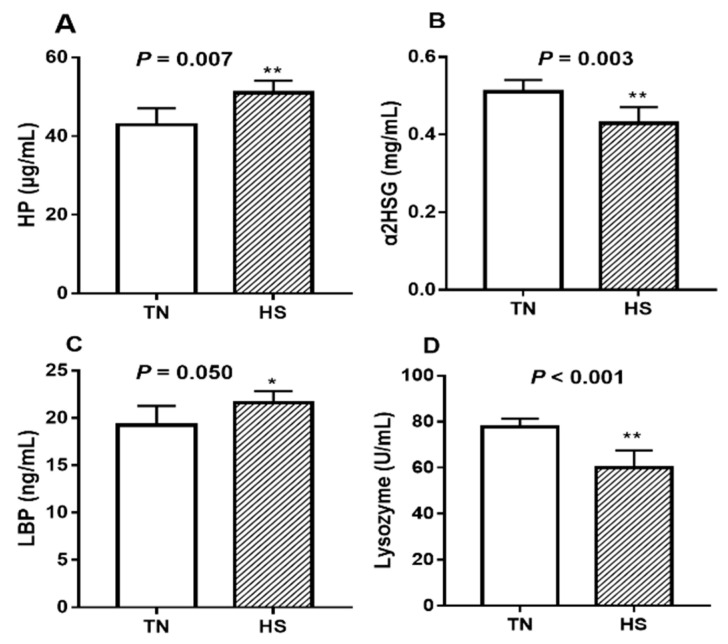
The effects of three weeks of thermal neutral (TN) or heat stress (HS) conditions on activities of (**A**) HP, (**B**) α2HGS, (**C**) LBP, and (**D**) lysozyme in the serum of finishing pigs. Data are mean ± SD; n = 8 for each group. * *p* < 0.05 and ** *p* < 0.01 before vs. after heat stress.

**Figure 3 animals-09-00395-f003:**
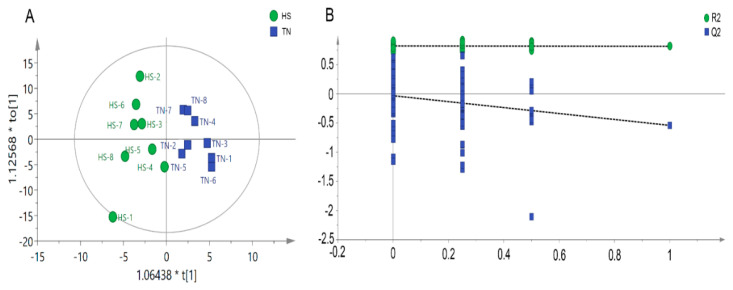
Score plot of orthogonal projections to latent structures discriminant analyses (OPLS-DA) derived from the GC-MS profiles of serum samples obtained from the heat stress (HS) group vs. thermal neutral (TN) group (**A**); Validation plots of the partial least squares discriminant analysis (OPLS-DA) models acquired through 999 permutation tests for the GC-MS data of serum metabolome (**B**).

**Figure 4 animals-09-00395-f004:**
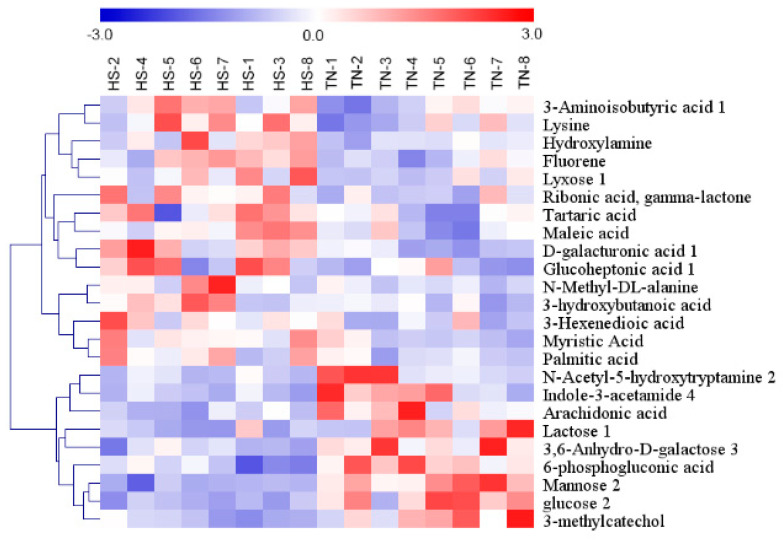
Hierarchical cluster analysis of twenty-four metabolites with significantly different levels in serum of finishing pigs. Red indicates high relative abundance, and blue indicates low relative abundance.

**Figure 5 animals-09-00395-f005:**
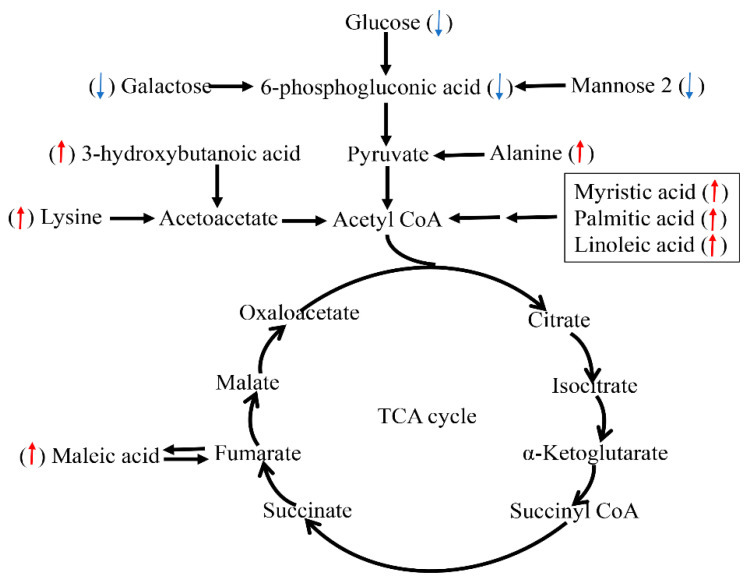
Schematic overview of some important metabolites and major metabolic pathways related to fat acid, amino acid, and energy metabolism in heat-stressed pigs. Red up-arrow: heat stress (HS) group vs. thermal neutral (TN) group up-regulation. Blue down-arrow: HS group vs. TN group down-regulation.

**Table 1 animals-09-00395-t001:** The effects of three weeks of thermal neutral (TN) or heat stress (HS) conditions on whole blood erythrocyte, hematocrit, hemoglobin, MCV, and MCH in the finishing pigs.

Parameters	TN	HS	*p*
Erythrocyte (×10^12^/L)	7.64 ± 0.093	7.583 ± 0.148 *	0.030
Hematocrit (%)	48.92 ± 0.85	44.37 ± 1.28 **	0.006
Hemoglobin (g/L)	138.92 ± 2.82	127.6 ± 3.42 *	0.010
MCV	64.06 ± 1.08	58.5 ± 1.20 **	0.004
MCH	18.18 ± 0.32	16.85 ± 0.36 **	0.032

MCV: mean corpuscular volume; MCH: mean corpuscular hemoglobin. Data are mean ± SD; n = 8 for each group. * *p* < 0.05 and ** *p* < 0.01 before vs. after heat stress.

**Table 2 animals-09-00395-t002:** Metabolites with significantly different levels in the serum of thermal neutral (TN) or heat stress (HS) finishing pigs.

NO.	Metabolites	RT	Quant Mass	VIP	*p*	Fold Change
Gut Microbiome-Derived Metabolites						
1	fluorene	9.41	204	2.60	0.009	1.51
2	lyxose 1	9.46	103	1.87	0.039	1.62
3	d-galacturonic acid	10.97	333	2.14	<0.001	1.76
Fatty Acid Metabolism						
4	3-Hexenedioic acid	8.65	170	1.91	0.049	2.10
5	myristic Acid	10.57	285	1.81	0.026	1.55
6	palmitic acid	11.56	117	1.96	0.043	1.55
7	arachidonic acid	13.02	80	1.16	0.034	0.76
8	3-Aminoisobutyric acid 1	8.29	174	1.36	0.020	1.23
9	3-hydroxybutanoic acid	5.91	131	2.10	0.037	1.87
10	maleic acid	7.18	341	1.70	0.017	1.84
11	tartaric acid	9.31	355	1.91	0.027	1.46
Carbohydrate Metabolism						
12	glucose 2	10.90	103	1.58	<0.001	0.52
13	mannose 2	10.82	160	2.23	<0.001	0.65
14	lactose 1	14.21	73	1.58	0.027	0.30
15	3,6-Anhydro-D-galactose	10.00	231	1.45	0.015	0.58
16	6-phosphogluconic acid	13.16	318	2.17	0.006	0.73
17	ribonic acid, gamma-lactone	9.70	68	2.31	0.024	1.53
18	Glucoheptonic acid	12.61	232	2.14	0.023	1.72
Amino Acid Metabolism						
19	lysine	10.95	174	1.39	0.015	1.71
20	N-Methyl-DL-alanine	6.25	130	1.88	0.050	2.19
Amines Metabolism						
21	hydroxylamine	5.88	245	1.62	0.031	1.56
22	N-Acetyl-5-hydroxytryptamine 2	13.57	55	1.95	0.049	0.15
23	3-methylcatechol	7.90	180	2.34	0.012	0.53
24	indole-3-acetamide	12.58	105	1.76	0.016	0.52

RT, retention time; VIP, variable importance in the projection; Fold Change, ratio of mean peak area of the HS group to the mean peak area of the TN group.

**Table 3 animals-09-00395-t003:** Partial Pearson’s correlation between APRs and metabolites with correction of the treatment groups (HS and TN).

Metabolites	HP	LBP	lysozyme	α2HSG
Fluorene	0.97 ***	NS	NS	NS
N-Acetyl-5-Hydroxytryptamine 2	NS	NS	0.96 ***	NS
Mannose 2	−0.51 *	−0.68 **	−0.52 *	−0.56 *
Indole-3-acetamide 4	NS	NS	0.92 ***	NS
Glucose 2	NS	−0.62 *	NS	−0.52 *
6 phosphogluconic acid	0.92 ***	NS	NS	NS
Arachidonic acid	0.91 ***	NS	NS	NS
3-Aminoisobutyric acid 1	NS	−0.62 **	−0.60 *	−0.55 *
Tartaric acid	NS	0.65 **	0.84 ***	NS
Hydroxylamine	NS	NS	0.93 ***	NS
Lysine	NS	−0.52 *	NS	NS
Maleic acid	NS	−0.58 *	NS	−0.51 *
Ribonic acid, gamma-lactone	NS	0.95 ***	NS	0.97 ***
Lyxose 1	0.53 *	0.83 ***	NS	0.86 ***
D-galacturonic acid 1	0.58 *	0.77 ***	NS	0.79 **
Glucoheptonic acid 1	NS	NS	0.65 **	NS
N-Methyl-DL-alanine	0.91 **	NS	NS	NS
3-Hexenedioic acid	NS	NS	NS	NS
3-hydroxybutanoic acid	NS	0.95 ***		0.97 ***
Myristic Acid	NS	NS	0.91 ***	NS
Palmitic acid	NS	0.92 ***	NS	0.95 ***
Lactose 1	NS	NS	NS	NS
3,6-Anhydro-D-Galactose 3	0.94 ***	NS	NS	NS
3-methylcatechol	NS	NS	NS	0.51 *

HP, haptoglobin; α2HSG, alpha-2-HS-Glycoprotein; LBP, LPS-blinding protein * *p* < 0.05, ** *p* < 0.01, and *** *p* < 0.001. NS, no significant correlations (*p* > 0.05).

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
