# Peer review of "Chronic Heat Stress Induces Acute Phase Responses and Serum Metabolome Changes in Finishing Pigs"

_animals, 2019, doi:10.3390/ani9070395_

Round 1

Reviewer 1 Report

The paper presents the results of a research on the effects of heat stress on pigs, identifying new possible marker molecules.

The research was conducted in a scientifically correct way and the paper is written in a clear and complete manner.

There are some typing errors that must be corrected before the paper can be accepted for publication:

line 83 metabolites and not metalolites

line 246 chronic and not chromic

line 256 erythrocytes and not erythrocyte

line 265 Hb and not HP

the label of figure 3, at the top right is not legible.

Author Response

To Reviewer 1:

The paper presents the results of a research on the effects of heat stress on pigs, identifying new possible marker molecules.

The research was conducted in a scientifically correct way and the paper is written in a clear and complete manner.

Response: Thank you for your helpful comments and suggestions.

There are some typing errors that must be corrected before the paper can be accepted for publication:

Response: We have corrected all grammatical and typing errors in the revised version.

1. line 83 metabolites and not metalolites

Response: We have corrected the error in revised version (line 80).

2. line 246 chronic and not chromic

Response: We have corrected the error in revised version (line 245)

3. line 256 erythrocytes and not erythrocyte

Response: We have corrected the error in revised version (line 245)

4. line 265 Hb and not HP

Response: Haptoglobin (HP), a positive APR, reduces the oxidative damage associated with hemolysis by binding free HP

5. the label of figure 3, at the top right is not legible.

Response: Thank you for your suggestions. We have replaced it with a legible figure in revised version.

Reviewer 2 Report

This paper reports the results of a serum metabolome analysis performed on heat stress treated pigs compared to a control group.

The written English is sometimes of poor quality. There are several mistakes, and I suggest the Authors ask for the help of an English native speaker. Several sentences are wrongly worded.

Some examples (but there are much more in the manuscript):

-line 30 “metabolites were significantly different levels” must be “metabolites had significantly different levels”

-line 36 “short-chain fatty acid” must be “short-chain fatty acids”

-line 47 “in most every livestock industry” makes no sense. Please reword the sentence.

-line 51-52: “For example, it is estimated that US$1 billion for swine is annually lost due to HS” is not clear what the authors are referring to. I suggest rewording the sentence.

Another issue is about the number of sampled animals. In line 91 the authors claim that they have 8 animals per group but in the following lines they report that 4 animals were moved to the artificial climate chamber. Please clarify this point.

On the whole anyway the article is of interest and quite well described. Another part I suggest the authors add is about humidity and pigs’ density in the artificial climate chamber, since these aspects could be important when dealing with heat stress.

Author Response

To Reviewer 2:

Comments and Suggestions for Responsethors

This paper reports the results of a serum metabolome analysis performed on heat stress treated pigs compared to a control group.

On the whole anyway the article is of interest and quite well described. Another part I suggest the Responsethors add is about humidity and pigs’ density in the artificial climate chamber, since these aspects could be important when dealing with heat stress.

Response: Thank you for your helpful comments and suggestions.

The written English is sometimes of poor quality. There are several mistakes, and I suggest the Responsethors ask for the help of an English native speaker. Several sentences are wrongly worded.

Response: Our manuscript has been edited by an English native speaker. We have checked it again and again to ensure absence of grammatical and typing errors in revised version of our manuscript.

Some examples (but there are much more in the manuscript):

1. line 30 “metabolites were significantly different levels” must be “metabolites had significantly different levels”

Response: We have revised it (line 30).

2. line 36 “short-chain fatty acid” must be “short-chain fatty acids”

Response: We have revised it (line 36).

3. line 47 “in most every livestock industry” makes no sense. Please reword the sentence.

Response: We have revised it (line 47).

4. line 51-52: “For example, it is estimated that US$1 billion for swine is annually lost due to HS” is not clear what the Responsethors are referring to. I suggest rewording the sentence.

Response: We have deleted it.

5. Another issue is about the number of sampled animals. In line 91 the Responsethors claim that they have animals per group but in the following lines they report that 4 animals were moved to the artificial climate chamber. Please clarify this point.

Response: Thank you for your suggestions. We have described it in details in revised version of our manuscript. (line 90-92).

Four pigs from one group were housed in an artificial climate chamber (2.1× 4.8 m 2, luminance 100 lx, photoperiod 14 h light, humidity 55% ± 5%) as described by our previous studies. Sixteen pigs were allocated to four the climate chambers.